# Reversibility in the Physical Properties of Agarose Gels following an Exchange in Solvent and Non-Solvent

**DOI:** 10.3390/polym16060811

**Published:** 2024-03-14

**Authors:** Denis C. D. Roux, François Caton, Isabelle Jeacomine, Guillaume Maîtrejean, Marguerite Rinaudo

**Affiliations:** 1Univ. Grenoble Alpes, CNRS, Grenoble INP (Institute of Engineering Univ. Grenoble Alpes), LRP, 38000 Grenoble, France; denis.roux@univ-grenoble-alpes.fr (D.C.D.R.); francois.caton@univ-grenoble-alpes.fr (F.C.); guillaume.maitrejean@univ-genoble-alpes.fr (G.M.); 2NMR Centers of RMN-ICMG (FR2607), CERMAV-CNRS, BP53, 38000 Grenoble, France; isabelle.jeacomine@cermav.cnrs.fr; 3Biomaterials Applications, 38000 Grenoble, France

**Keywords:** agarose, methyl substituent, solvent retention, rheology, gel, solvent exchange

## Abstract

Agarose forms a homogeneous thermoreversible gel in an aqueous solvent above a critical polymer concentration. Contrary to the prevailing consensus, recent confirmations indicate that agarose gels are also stable in non-solvents like acetone and ethanol. A previous study compared gel characterisations and behaviours in water and ethanol, discussing the gelation mechanism. In the current work, the ethanol gel is exchanged with water to explore the potential reversibility of the displacement of water in agarose. Initially, the structure is characterised using ^1^H NMR in DMSO-*d*_6_ and D_2_O solvents. Subsequently, a very low yield (0.04) of methyl substitution per agarobiose unit is determined. The different gels after stabilisation are characterised using rheology, and their physical properties are compared based on the solvent used. The bound water molecules, acting as plasticizers in aqueous medium, are likely removed during the exchange process with ethanol, resulting in a stronger and more fragile gel. Next, the gel obtained after the second exchange from ethanol back to water is compared with the initial gel prepared in water. This is the first time where such gel has been characterised without undergoing a phase transition when switching from a good solvent to a non-solvent, and vice versa, thereby testing the reversibility of the solvent exchange. Reversibility of this behaviour is demonstrated through swelling and rheology experiments. This study extends the application of agarose in chromatography and electrophoresis.

## 1. Introduction

Agarose is a linear polysaccharide composed of repeating units of agarobiose that is extracted from red algae in boiling water. The repeat unit is the following:→3) -β-D-galactose-(1→4) -3,6-anhydro-α-L-galactose-(1→

When agarose is dissolved in hot water, it takes a coiled conformation known as the “sol state” [1,2]. Depending on the molar mass, when the agarose concentration reaches a critical value of approximately 1 g/L, and when the temperature of the solution decreases, the gel state is achieved. The sol–gel transition is characterised by a large hysteresis loop observed in all material properties (RMN, rheology, …) during the cooling and heating steps [2]. Further, it is associated with a coil–helical conformation.

While single-chain helical conformations have occasionally been observed in molecular modelling [3] and X-ray diffraction on extended solid films [4] or concentrated gels [5,6], a double helical conformation made of two parallel chains stabilised by a H-bond network is favoured [5,6,7,8]. Then, the gelation process involves the formation of double helices which associate in rigid bundles, creating a physical network structure and producing a turbid gel [6]. It has been demonstrated that, in the gel state, the degree of order increases with polymer concentration. This correlates with the stacking of double helices into crystalline domains [5].

The interaction with water molecules is claimed to stabilise this microstructure [7,8]. But, through Monte Carlo simulations, Corongiu et al. concluded that there was no water in the cavity formed by the double helices [9]. Furthermore, NMR relaxation investigations show that there is a rapid exchange of bound and free water molecules when agarose is present [5]. An analogy between agarose and carrageenan has sometimes been drawn. However, since carrageenan is not neutral, it is necessary to screen electrostatic repulsion sites between the chains using excess KCl [8,10].

Recently, we proposed a model [10,11] that demonstrates the ability of kinetics and slow syneresis to account for gelation without the need to invoke kinks or defects in the molecular structure for connections, as proposed historically [8,12,13]. This syneresis phenomenon is attributed to the contraction of the polymer network through the slow and continuous aggregation of helices and the release of water from the gel [5,8].

The origin and purification of agarose affect its chemical structure. In particular, the -OH sites can be substituted. The substituents are often identified using NMR [14,15]. According to the molecular structure and depending on the molecular weight, the mechanical response of agarose differs [16,17,18,19]. Some experiments conducted using mechanical tension/compression and oscillatory shear rheology, involving three different agarose molecular weights, concluded that the stress rupture, shear modulus, and Young modulus depend on both concentration and molecular weight [18]. It is recognised that, for all agarose gels, elasticity is a consequence of thick connected bundles that form large pores [19]. The interconnected large pores allow the use of agarose for chromatography or electrophoresis of large DNA molecules [20].

At low concentrations, agarose gels are soft, while at high concentrations, they become more rigid. Both types of gels, within the linear viscoelastic domain, display significant thermoreversible hysteresis, extensively documented in the literature and interpreted as a consequence of coil–helix transition associated with the sol–gel transition [2]. Indeed, agarose gels exhibit heterogeneity on a small scale due to the presence of bundles of aggregated double helices [6]. This structural heterogeneity is influenced by the thermal history of the samples [21]. For the gel system, a flat linear behaviour in the moduli G′ and G″ as a function of oscillation strain is observed, followed by a sharp decrease in G′ at high strain and a small overshoot in G″ [22]. This behaviour is interpreted by these authors as a consequence of a break of an interconnected network of cross-linked gel particles. The observation of G″ overshoκot is more pronounced at high agarose concentrations [23]. 

Unlike chemical gels with covalent bonds, agarose is a physical gel based on non-covalent interactions, forming a 3D interconnected network. According to rheology experiments on stress relaxation, the apparent activation energy of the agarose gels is approximately 5 kcal/mol, which is of the same order of magnitude as that of hydrogen bond breaking. Those data confirm that the hydrogen bonding plays a major role in the gelation mechanism of agarose gels [24].

As typically observed, a phase transition occurs when swollen gels in a good solvent are immersed in a non-solvent of the considered polymers [25,26]. However, it has been shown that physical gels, such as agarose and κ-carrageenan formed in water, can be also obtained by exchange with a miscible non-solvent. In this case, the swelling ratio is almost independent of the solvent/non-solvent ratio [27]. This was recently confirmed for the agarose–water–ethanol system [2], where it was shown that the ethanol-swollen agarose gel is stiffer than in water, even if all water molecules have been displaced from agarose. 

In contrast to the traditional constant-gap procedure, following Mao et al. [28,29], in agreement with Ewoldt et al. [30], and recognizing the challenge in accurately characterizing the rheological parameters of agarose gels, it is necessary to apply a controlled normal force during the temperature ramp. Furthermore, these authors found no difference in viscoelastic values in the linear oscillation when they used either a smooth or a rough surface.

Finally, it has been recently demonstrated that upon exchanging water with ethanol, a stable gel is obtained, which is no longer thermoreversible up to 80 °C and stiffer with a higher G′ modulus [2]. From this work, it was concluded that the H-bond network is reinforced by the displacement of water molecules. The objective of the present work is to test for the reversibility of water molecule interactions with agarose in an aqueous medium and to compare the rheological behaviour of different gels stabilised in water and ethanol at 25 °C.

## 2. Materials and Methods

### 2.1. Agarose Samples Preparation

Agarose was provided by Genetics, Nippon Genetics Europe.Cat.AG01 (Düren, Germany) with a high melting temperature (T_f_ ~ 80 °C). It was used without purification. For rheology, samples were prepared by dissolving agarose in deionised water maintained at 85 °C for 1 h. NMR characterization was performed on agarose directly solubilised in DMSO-*d*_6_ and D_2_O at a concentration of 10 g/L and a temperature of 80 °C. This concentration is well above the critical gel formation concentration which is 1 g/L.

For the rheology test, disk samples with a diameter of 25 mm were prepared by gently pouring a hot solution at a concentration of 10 g/L into a cylindrical mould. The temperature was lowered by keeping the mould at an ambient temperature for half a day to obtain a stable gel. The thickness of the cylindrical samples was controlled by the total volume poured into the mould, resulting in disk samples of thicknesses between 3.5 and 4.5 mm. The samples were then stored at 5 °C and used within a few days. Under the conditions tested, particularly with an agarose concentration of 10 mg/mL, no syneresis was observed during the experiments. 

The initial gel samples formed in water were directly obtained from the dissolution of agarose in water at high temperature and cooling, referred to as H_2_O_initial_ in the following text, denoting the initial sample. Secondly, EtOH samples were prepared by taking water gel samples and directly immersing them in ethanol, with the ethanol being replaced each day for a total of four days in order to exchange the water with ethanol. In the following, these samples are referred as EtOH. Finally, EtOH samples were taken and an ethanol-to-water exchange was obtained by immersing EtOH samples in water and replacing the water each day for four days. These samples were then referred to as H_2_O_final_ samples, denoting the re-exchange samples.

### 2.2. Water Regain and Swelling Degree

The degree of swelling was determined by the weight of swollen gel (W_h_) in the solvent of interest, either H_2_O or ethanol, and the dried weight (W_s_) expressed in mL of solvent/g of dried gel, taking into account the ethanol density (d = 0.79). The dried weight was obtained after 2 h at 120 °C.

### 2.3. Rheology

Rheology was performed on an ARES-G2 rotational rheometer (TA Instruments, New Castle, DE, USA) equipped with plate geometries of 25 mm in diameter (Figure 1). Two stainless-steel plate–plate geometries were used, one with a smooth surface and a second one with a rough surface formed of side-by-side pyramids, each with 0.1 mm sides and 0.1 mm in height. Temperature control at 25 °C was ensured by an Advanced Peltier System (TA Instruments). To prevent evaporation, an in-house ring system allowed cylindrical gels positioned between the top and bottom plates to be immersed or not in a bath of water or ethanol (Figure 1a). As expected, no influence of the presence of submerged fluids was observed on the torque measured in the measurement range. Based on this observation, all experiments presented in this article were conducted with the appropriate solvent, either water or ethanol. In order to slow down the solvent’s evaporation, a conventional plastic cover was also used.

Experiments were conducted with particular attention paid to the contact between the gel and the plates. To achieve this, a small axial force of 0.10 ± 0.01 N was maintained perpendicular to the upper plate surface. Two sets of tools were used: one with a pair of smooth plates and the other with rough plates. The smooth tools are shown in the photo in Figure 1c and the top plate of the rough tools is shown in the photo in Figure 1b.

## 3. Results and Discussion

### 3.1. NMR Study

Analysis of the agarose sample is performed using 1D and 2D ^1^H and ^13^C NMR spectroscopies. The spectra are compared with those obtained previously on the agarose sample from Sigma chemicals [2] and referring to the chemical structure given in Figure 2. In this work, the discussion is limited to proton NMR spectroscopy, because carbon spectra are similar for the two samples. 

The proton NMR spectra in D_2_O and DMSO-*d*_6_ are presented in Figure 3 and Figure 4, respectively. The spectra are provided for the polymer studied previously [2] and the actual sample.

The comparison shows that, in the current sample (Figure 3b), there is a significantly smaller signal corresponding to the methyl substituent at 3.43 ppm than in the Sigma sample (Figure 3a). The triplet for H-2′ is now separated from other signals due to the absence of the H-6″ signal. 

In DMSO, the methyl group signal is very small and observed at 3.3 ppm, while from 3.4 to 3.8 ppm, the spectrum is simplified due to the absence of -OH-6″ and H-6″ (Figure 4a,b). All the chemical shifts for the first sample are given in Appendix A with a modification for H-4 at 4.54 ppm and not 4.34 as indicated in our previous paper [2]. Except for this change and the suppression of the methylated galactose (G″) unit, all the chemical shifts for anhydro-L-galactose (G) and D-galactose (G′) units given in Appendix A remain unchanged for the Genetics sample (see Appendix A).

### 3.2. Degree of Substitution on Agarose

From the spectrum obtained in DMSO-*d*_6_, it is shown that the proton with a narrow signal at 3.3 ppm is assigned to a methyl substituent correlated with the C-6 of the D-galactose unit. Similarly, the carbon from the methyl group (not shown) correlates with the proton H-6 of D-galactose. Considering the integral of this signal in reference to the H-1 of anhydro-L-galactose, it comes to DS = 0.04 ± 0.01, indicating a very low degree of methylation of D-galactose units. This value is confirmed from the spectrum in D_2_O. This result allows one to conclude that the selected sample may be considered a model for pure agarobiose polymer. The modification of the NMR spectra in both solvents is in agreement with the decrease in the degree of methylation. 

### 3.3. Degree of Swelling

The first step was to determine the degree of swelling of the gel in water and after the exchange in the presence of ethanol and then after the exchange back to water. It has been shown from DSC that in ethanol, the exchange of water molecules is complete [2]. This supports the idea that interacting water molecules are located outside the agarose’s inner cavity [2,9]. The experiments were performed after one month of stabilisation in a large amount of each solvent. In water, the initial degree of solvation is (W_h_ − W_s_)/W_s_ = 92 ± 0.5 g/g or mL/g dried gel in agreement with the weight concentration of the solution prepared at 10 g/L, which should give 100 as found in our previous paper [2]. This slightly lower value indicates that, after one month, there is a minor effect of syneresis. This effect is slow due to the relatively high agarose concentration and can be ignored during the period of rheological experiments.

In ethanol, the solvent content is given by (W_h_ − W_s_) × 0.79/W_s_ = 83 ± 0.5 mL/g dried gel indicating that the volume change is small when comparing the two solvents, confirming our previous results. After a second exchange to water, the degree of swelling is again 92 g/g dried agarose, proving the reversibility of the exchange. 

It is concluded that the degree of swelling (and consequently the porosity) of agarose at 10 g/L in water and ethanol is nearly the same, confirming our previous results [2], even if ethanol is a non-solvent of agarose. Hence, during the process, ethanol replaces water in the pores, inducing very little shrinkage. These results are in agreement with results obtained for aqueous agarose gel exchanged with acetone in which the length of a piece of gel (L_o_) decreases (L) as L/L_o_ ~ 0.93 in agarose [31].

### 3.4. Rheology of Agarose Gels

The gels corresponding to the three states of solvent H_2_O_initial_, EtOH, and H_2_O_final_ were studied to elucidate the modification of the rheological properties of the gels in the solvent and non-solvent media as well as to examine the reversibility of the solvent exchange. 

#### 3.4.1. Selection of Experimental Conditions

To ensure contact between the plates and the sample, a small normal force of 0.10 ± 0.01 N was imposed on the sample after loading, preventing a de-bonding of the sample between the plate–plate geometry when performing the measurements. To compare rheological experiments made with smooth and rough surfaces, all experiments were performed at the same controlled normal force (0.10 ± 0.01 N) applied onto immersed samples.

In Figure 5, the evolution of elastic and dissipation moduli versus the strain at a constant frequency of 1 Hz is depicted for an agarose gel of 10 mg/mL, using both smooth and rough plate–plate geometries. Figure 5a,b reveals a linear domain below the vertical dashed lines for both water gel and ethanol gel. The linear domain of agarose water gel, for strain below 0.2–0.3%, agrees with our previous work [2] and is in agreement with Normand [18]. The linear domain in strain of the ethanol gel (~0.09%) is about four times lower than for the water gel (~0.3%) regardless of the surface used in contact with the gels, smooth or rough. 

Over the limit of linearity, for both gels, regardless the contact surface used, elastic and dissipative moduli follow a non-linear behaviour characterised by a continuous but significant decrease in the elastic modulus and an overshoot in the dissipative modulus.

In summary, a study of the different gels’ behaviour as a function of the frequency is performed on immersed samples with their solvent, to prevent evaporation. A constant normal force of 0.10 ± 0.01 N is imposed on the gel samples, to ensure contact between the gel and the rough plate–plate geometry is selected. Finally, a constant strain of 0.01% is maintained to study the gels in the linear regime. 

#### 3.4.2. Viscoelastic Moduli of the Gels in the Linear Domain

When the exchange with ethanol was completed, it considerably reduced the range of linearity of the initial water gel from 0.3% to 0.09% as shown in Figure 5. In this range at 1 Hz, elastic G′ and dissipative G″ moduli are larger in ethanol gel than in water gel. For both gels, the ratio G′/G″ are large with a value of 30 for water gel and 15 for the ethanol gel. Moreover, the elastic modulus of ethanol gel (G′) is found to be four times that of the water gel. These results confirm that the ethanol gel is more rigid than the water gel and corroborate the suppression of the water during the exchange of water to ethanol. We interpret the increase in rigidity as an increase in H-bond formation between double helices and bundles. These results confirm our previous observations [2].

#### 3.4.3. Evidence of Ethanol–Water Reversibility

After selecting the optimal conditions to obtain valuable gel characteristics, the three states of the agarose gel are compared below.

Figure 6 shows the linear viscoelastic measurements conducted on the three gels: initial (H_2_O_initial_), exchange with ethanol (EtOH), and re-exchange from ethanol to water (H_2_O_final_). All samples exhibit a gel behaviour where the elastic moduli are independent of the frequency and are at least an order of magnitude higher than the viscous modulus. Whatever the frequency, moduli ratios G′/G″ are still around 30 in water and 15 in ethanol, confirming the plasticiser role of water. In addition, G′ in ethanol is 4 times larger than in water, confirming that higher rigidity of ethanol gel. Surprisingly, the degree of swelling is nearly the same in both media, indicating similar porosity and control of rigidity by interacting water.

Remarkably, the water-exchanged gel from EtOH gel has moduli close to those of the initial water gel. The slight difference between the final and initial water gels is comparable to the reproducibility of our experiments. These data allow to conclude that the solvent–non-solvent exchange is reversible at ambient temperature, only if the considered solvents are miscible.

## 4. Conclusions

This study focuses on an agarose sample with a very low degree of methylation (DM = 0.04) as determined by proton NMR spectroscopy. The objective is to investigate the impact of water–ethanol exchange on swelling and the rheology of agarose gel (prepared at 10 g/L in water).

Initially, the experimental conditions for rheological experiments are explored to ensure reproducible and significant results. A protocol is proposed: the gel sample is placed between both rough and smooth plate–plate geometries immersed in the solvent. Then, a small normal force (F_N_ = 0.1N) is applied during the strain sweep at a fixed frequency to ensure proper contact between the gel and the geometry. Under these conditions, reproducible and valuable parameters are obtained with both geometries. However, the rough geometry is preferred to prevent any slipping and a strain of 0.01% is chosen to ensure being in a linear domain regardless the solvent considered.

In the second part, the rheological parameters (G′ and G″) are determined and compared for three states of the agarose gel: H_2_O_initial_, EtOH, and H_2_O_final_. The results show the characteristic behaviour of the physical gel with constant elastic and viscous moduli as a function of the frequency. Immersed in ethanol, the degree of swelling and porosity are preserved, but the gel modulus G′ is much higher (4 times) than that of water gel. This is due to the removal of water molecules between double helices, which have a plasticising effect [2].

Ultimately, when ethanol is replaced by water, the mechanical properties of the gel in water are restored, and the hydrogen bonds of water molecules regain their plasticizer effect in the assembly of double helices within bundles. The H_2_O_final_ obtained after ethanol–water exchange exhibits the same degree of swelling as observed initially, and the G′ modulus obtained for H_2_O_initial_ is recovered.

This leads to the conclusion that the solvent interaction is reversible, as demonstrated here for the first time.

It is evident that the mechanical properties of agarose gels are influenced by the nature of the solvent/non-solvent and can be modified by a simple solvent exchange. This occurs only when the non-solvent miscible with water. This original behaviour is significant, paving the way for the anticipated use of those porous gels in chromatography and/or electrophoresis, irrespectively of the solvent/non-solvent medium, as long as they are miscible.

## Figures and Tables

**Figure 1 polymers-16-00811-f001:**
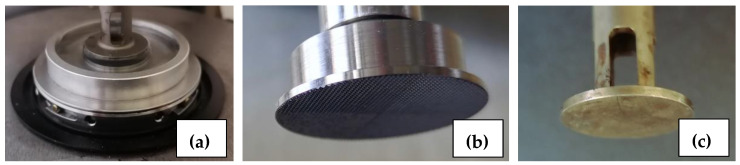
Photographs of plate–plate tools: (**a**) plate–plate tools with the immersed bath located at the lower plate position, (**b**) view of one of the rough plates constituted by side-by-side pyramidal geometries, and (**c**) view of one of the smooth geometries. All plates have a diameter of 25 mm.

**Figure 2 polymers-16-00811-f002:**
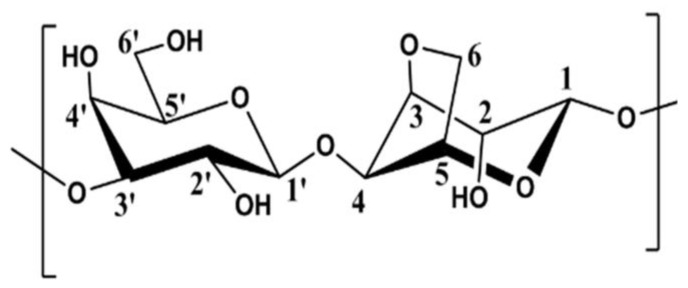
Repeat unit of agarose in which C′_6_ position of D-galactose is partially methylated and named C″_6_. The numbers shown in the structural diagram refer to the position of each carbon and correspond to the peaks assigned in the NMR spectra.

**Figure 3 polymers-16-00811-f003:**
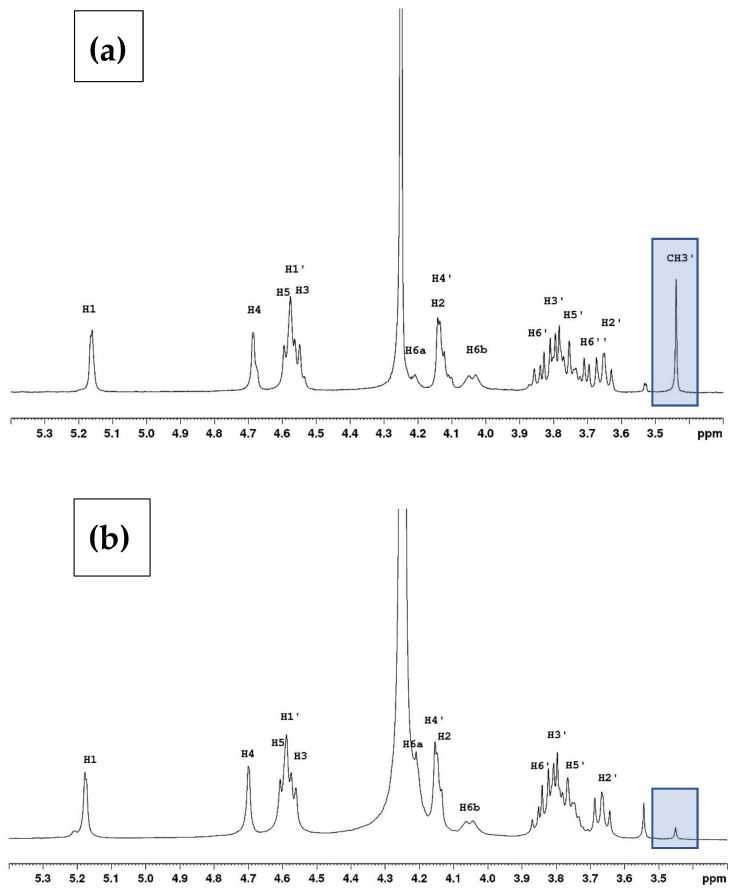
Proton NMR spectra of agarose in D_2_O at 80 °C with two degrees of methylation (DM) identified on the spectra. (**a**) From reference [2] with a degree of methylation (DM = 0.24) from Sigma and (**b**) with the new sample from Genetics (DM = 0.04). The tests were conducted a minimum of three times on each type of sample.

**Figure 4 polymers-16-00811-f004:**
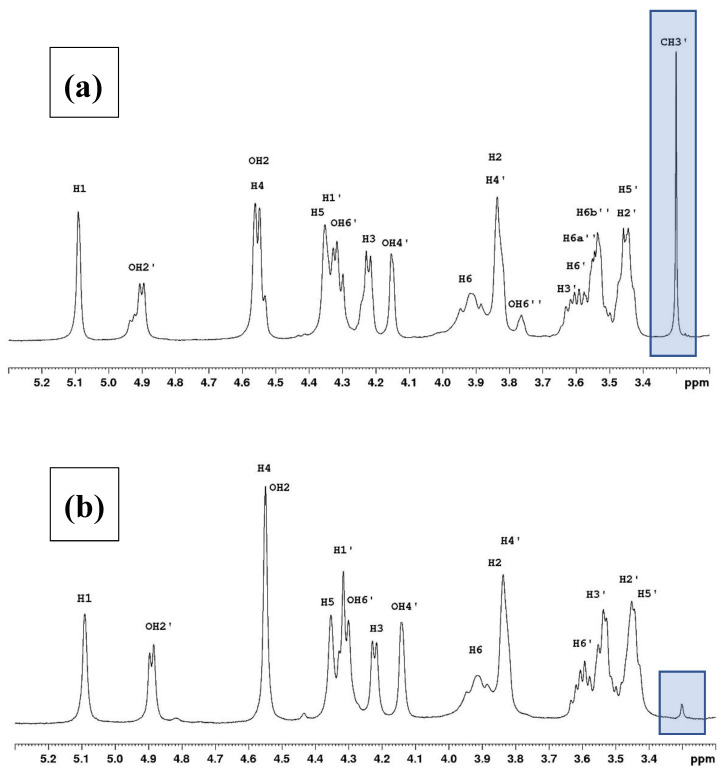
Proton NMR spectra in DMSO-*d*_6_ at 80 °C with two degrees of substitution identified in the spectra: (**a**) from reference [2] with a degree of methylation from Sigma (DM = 0.24) and (**b**) with the new sample from Genetics (DM = 0.04).

**Figure 5 polymers-16-00811-f005:**
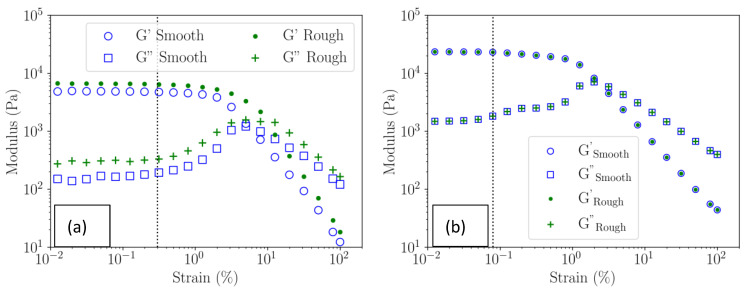
Elastic (G′) and dissipation (G″) moduli as a function of strain for 10 g/L agarose gels at a frequency of 1 Hz, measured for a gel with water solvent (**a**) and a gel with ethanol solvent (**b**). The blue symbols were obtained using a smooth plate geometry, and the green ones using a rough plate geometry. Both samples were measured at T = 25 °C. The linear domains of the water gel and ethanol gel occur for values below the vertical dotted line.

**Figure 6 polymers-16-00811-f006:**
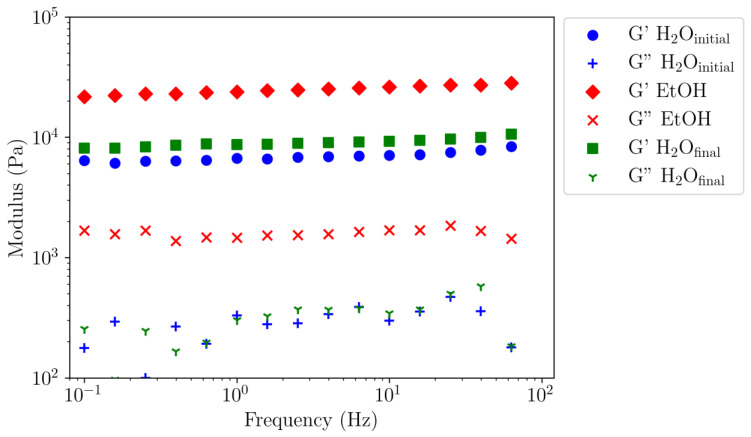
Elastic (G′) and dissipative (G″) moduli as a function of frequency at 0.01% strain for 10 g/L agarose gels under different solvent conditions at 25 °C. Blue symbols correspond to the initial gel in water (H_2_O_initial_). Green symbols correspond to the gel exchange with ethanol and the red symbols correspond to the gel re-exchange from ethanol to water (H_2_O_final_).

## Data Availability

Data are contained within the article and Appendix A.

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
