# Peer review of "Reversibility in the Physical Properties of Agarose Gels following an Exchange in Solvent and Non-Solvent"

_polymers, 2024, doi:10.3390/polym16060811_

Round 1

Reviewer 1 Report

Comments and Suggestions for Authors

Agarose is capable of forming gels in water under certain conditions. Water in such a gel can be replaced with alcohol, followed by reverse substitution. The authors of the presented manuscript study the properties of such systems. First of all, I was interested in the rheological behavior of these systems.

The abstract is well written and reflects the meaning of the manuscript. In the list of keywords, I suggest that authors replace the following keywords "gel in water and ethanol; reversibility of solvent exchange".
The methodological part fully reflects the algorithms for preparing gels in various media (water and alcohol) and a description of the research methods used.

L. 36, 37. It is not clear what kind of dependence the authors are talking about?
L. 122. “The H2O samples...” - I recommend rephrasing the introductory part.
Figures 1 and 2 can be deleted.
L. 290. "10g/Lin" you need to add a space.

The presented manuscript is easy to read and executed at an excellent level from the standpoint of studying the rheological properties of gels. It might not be a bad idea to add viscosity dependencies, as well as data for systems with partial replacement of water by alcohol. The conclusions of the manuscript reflect all the results achieved and the solution to the problem. As a recommendation, the authors should check the text and correct minor errors. Also, adding morphological data would enhance the work.

Reviewer 2 Report

Comments and Suggestions for Authors

See attached file.

Comments on the Quality of English Language
